# *Legionella pneumophila* Infection of Human Macrophages Retains Golgi Structure but Reduces O-Glycans

**DOI:** 10.3390/pathogens11080908

**Published:** 2022-08-12

**Authors:** Yanlin Fu, Vinitha Macwan, Rebecca Emily-Sue Heineman, Mauricio R. Terebiznik, Rene E. Harrison

**Affiliations:** Department of Cell & Systems Biology and the Department of Biological Sciences, University of Toronto Scarborough, Toronto, ON M1C 1A4, Canada

**Keywords:** macrophage, *Legionella*, Golgi body, glycosylation

## Abstract

*Legionella pneumophila* is an accidental pathogen that replicates intracellularly within the *Legionella*-containing vacuole (LCV) in macrophages. Within an hour of infection, *L. pneumophila* secretes effectors to manipulate Rab1 and intercept ER-derived vesicles to the LCV. The downstream consequences of interrupted ER trafficking on the Golgi of macrophages are not clear. We examined the Golgi structure and function in *L. pneumophila*-infected human U937 macrophages. Intriguingly, the size of the Golgi in infected macrophages remained similar to uninfected macrophages. Furthermore, TEM analysis also did not reveal any significant changes in the ultrastructure of the Golgi in *L. pneumophila*-infected cells. Drug-induced Golgi disruption impacted bacterial replication in human macrophages, suggesting that an intact organelle is important for bacteria growth. To probe for Golgi functionality after *L. pneumophila* infection, we assayed glycosylation levels using fluorescent lectins. Golgi O-glycosylation levels, visualized by the fluorescent cis-Golgi lectin, *Helix pomatia* agglutinin (HPA), significantly decreased over time as infection progressed, compared to control cells. N-glycosylation levels in the Golgi, as measured by L-PHA lectin staining, were not impacted by *L. pneumophila* infection. To understand the mechanism of reduced O-glycans in the Golgi we monitored UDP-GalNAc transporter levels in infected macrophages. The solute carrier family 35 membrane A2 (SLC35A2) protein levels were significantly reduced in *L. pneumophila*-infected U937 and HeLa cells and *L. pneumophila* growth in human macrophages benefitted from GalNAc supplementation. The pronounced reduction in Golgi HPA levels was dependent on the translocation apparatus DotA expression in bacteria and occurred in a ubiquitin-independent manner. Thus, *L. pneumophila* infection of human macrophages maintains and requires an intact host Golgi ultrastructure despite known interference of ER–Golgi trafficking. Finally, *L. pneumophila* infection blocks the formation of O-linked glycans and reduces SLC35A2 protein levels in infected human macrophages.

## 1. Introduction

*Legionella* is a Gram-negative bacteria that forms biofilms and infects amebae in natural or manmade fresh water systems. *Legionella* has more than 60 species and about 70 serogroups [1]. *Legionella*
*pneumophila* (*L. pneumophila*) serogroup 1 is the most common type and can cause Legionnaires’ disease in human, which is a type of pneumonia that can lead to death in elderly patients with pre-existing lung conditions as well as immunocompromised and immunosuppressed patients [2]. After *L. pneumophila*-containing aerosols are inhaled into the human lung, *L. pneumophila* is phagocytosed by alveolar macrophages into a *Legionella*-containing vacuole (LCV). Upon internalization, the bacteria deliver effector proteins via a Dot/Icm type 4B secretion system to modulate a series of host cell trafficking pathways and inhibit the lysosome proton pump to evade destruction by the host cell lysosomes [3]. Bacteria effector proteins also co-opt the host cell energy supply to allow replication within the cells and eventually apoptosis to release replicated *L. pneumophila* into the extracellular space [4].

The recruitment of ER-like vesicles to the LCV membrane is observed during the first 15–30 min of infection of U937 macrophages [5]. To redirect ER membrane trafficking, *L. pneumophila* hijack and recruit Rab1, an important regulator of vesicle trafficking between the ER and Golgi, to the LCV [6,7]. This is induced by the early effector SidM, which is anchored to the LCV via a PI(4)P binding domain and sequesters and maintains Rab1 in an active form [8,9,10]. Post-translational AMPylation of Rab1 activates a subset of Rab1-dependent signaling pathways, causing the tethering of ER-derived vesicles to the LCV [11,12]. SidM and Rab1 levels on the LCV start to decrease 2 h after infection in macrophages [13]. *L. pneumophila* then utilizes SidD as an antagonistic effector to efficiently deAMPylate Rab1 [12,13]. In addition, it translocates its own Rab1 GAP, the effector LepB, in order to have complete control of Rab1 function. This effector induces GTP hydrolysis and allows Rab1 to be susceptible to membrane extraction by GDIs [13]. The bacterial effector AnkX also phosphocholinates any inactive Rab1, preventing its extraction from the LCV membrane [14,15]. In addition to Rab1, *L. pneumophila* also targets Arf1, which is a small GTPase that is involved in regulating budding and uncoating of vesicles in the ER–Golgi intermediate compartment and Golgi [16]. A T4SS effector RalF activates Arf1 and recruits it to the LCV via its Sec7-homology domain [17], thus promoting efficient fusion of ER-derived vesicles to the LCV [7,18]. Recent work has identified more ER resident protein and Rab targets of *L. pneumophila* effectors to promote ER vesicle budding and efficient targeting to the LCV [7,19].

Compared to the wealth of evidence demonstrating that *L. pneumophila* markedly disrupts ER trafficking pathways, much less is known about its effect on the Golgi complex. The Golgi is composed of flattened, disk-shaped, intracellular membrane containing structures called cisternae which are organized into stacks [20]. Golgi cisternae are divided into three parts, with ER-derived vesicles merging with cis-Golgi and cargo proteins continuing travel from cis- to medial- and to trans-Golgi to undergo post-translational modifications such as N-glycosylation, O-glycosylation, and S-palmitoylation. From the trans-Golgi the modified proteins are sent to their destinations including lysosomes and the plasma membrane or are secreted into extracellular space [21]. Most glycoproteins are N-glycosylated, which starts with the addition of *N*-acetylglucosamine (GlcNAc) to the nitrogen of an asparagine residue. N-glycans are categorized into three types: high mannose, hybrid, and complex N-glycans. High mannose N-glycans are attached to glycoproteins in the ER while hybrid and complex N-glycans are formed on glycoproteins in Golgi [22]. O-glycosylation occurs exclusively in the Golgi and starts with the addition of GalNAc from UDP-GalNAc to serine or threonine residues on proteins. This process is catalyzed by polypeptide-N-acetyl-galactosaminyltransferases in the Golgi [23].

The purpose of our study was to determine whether known ER disruptions caused by *L. pneumophila* had downstream impacts on Golgi structure and function in macrophages. To understand Golgi impacts caused by *L. pneumophila*, we quantitatively assessed the Golgi volume and imaged its ultrastructure in infected human macrophages. We experimentally disrupted the Golgi structure and monitored *L. pneumophila* growth in macrophages. Next, we assessed the impact of *L. pneumophila* infection of glycan production in the Golgi, using fluorescently tagged lectins and image analysis. Finally, we measured levels of a key Golgi sugar transporter in infected cells and assayed the benefit of sugar supplementation on *L. pneumophila* growth.

## 2. Materials and Methods

### 2.1. Reagents and Antibodies

U937 human monocytic cells (CRL-1593.2) and HeLa cells (CRM-CCL-2) were obtained from ATCC. RPMI 1640 and fetal bovine serum was from Gibco (Gaithersburg, MD, USA). The rabbit polyclonal SLC35A2 antibodies were from Abcam (ab222854) (Cambridge, UK) and Novus biologicals (NBP3-10880) (Littleton, CO, USA), the mouse monoclonal GM130 antibody was from BD Biosciences (610823) (Mississauga, ON, Canada), rabbit polyclonal TGN46 antibody was from Abcam (ab50595), the mouse monoclonal LP1 Legionella antibody was from EMD Millipore Canada (MAB10223) (Etobicoke, ON, Canada), the LAMP1 antibody (ab24170) was from Abcam, and the Alexa Fluor^TM^488 conjugated HPA lectin and PHA-L lectin were from Life Technologies (L11271 and L11270, respectively) (Carlsbad, CA, USA). Paraformaldehyde was from Electron Microscopy Sciences (Hatfield, PA, USA). WT, *L. pneumophila* strain 02 (*Lp*02) and the rabbit polyclonal anti-*Legionella* antibody were from Dr. Mauricio Terebiznik’s laboratory (University of Toronto Scarborough, Scarborough, ON, Canada). The *Legionella pneumophila* strain *Lp*02 was used in this study [24]. The ΔDotA *Lp*02 strain was originally obtained from Dr. R Isberg (Tufts University Medical School, Boston, MA, USA) and characterized in [25,26]. The ΔAnkB mutant strain (*Lp*02 Δ*lpg* 2144 or Δ*legAU13*) was a gift from Alexander Ensminger (University of Toronto, Scarborough, ON, Canada) [27]. N-Acetyl-D-galactosamine (GalNAc, A2795), MG132 (M7449), and all other reagents and tubulin and actin antibodies were from Sigma-Aldrich (Oakville, ON, Canada).

### 2.2. Cell Culture

U937 cells in suspension were maintained at 37 °C in an incubator supplied with 5% CO_2_ in RPMI 1640 with 10% heat inactivated FBS. Cells were passaged every 2 or 3 days to maintain cells at a cell density of 1 × 10^5^ to 2 × 10^6^ cells/mL in T75 tissue culture flasks. Then, 5 × 10^5^ U937 cells were passaged into 12-well plates in RPMI with 10% FBS supplemented with 1 μM PMA for 24 h to differentiate into monolayer macrophages. HeLa cells were grown in DMEM containing 10% FBS and passaged as described [28].

### 2.3. L. pneumophila Growth and Cell Infections and Treatments

Bacteria from frozen glycerol stocks were streaked onto buffered charcoal yeast extract (BCYE) agar plates containing ACES. The ΔDotA and ΔAnkB *L. pneumophila* mutant strains were streaked onto BCYE plates supplemented with 100 ug/mL thymidine. To obtain the short rod form of *Legionella*, bacteria were scraped from BCYE plates after growth for 3 days and cultured in BYE broth for 24 h at 37 °C with shaking at 100 rpm. To further enrich growth of short rods, this culture was sub-cultured in fresh BYE broth supplemented with an additional amount (50%) of α-ketoglutarate reagent at an OD_600_ of 0.05 and grown at 37 °C with shaking at 100 rpm to an OD_600_ of 2.0–3.0.

For infections, the *L. pneumophila* strain Lp01 at an OD_600_ of 2.0–3.0 was added onto differentiated U937 macrophages or HeLa cells with a Multiplicity of Infection (MOI) of 50 or 100. Plates were spun at 4 °C for 5 min at 300× *g* and cells incubated further at 4 °C for 5 min to promote bacterial attachment. Cells were next incubated at 37 °C for 30 min followed by three 1X PBS washes and addition of fresh RPMI media. Infections were allowed to proceed at 37 °C with 5% CO_2_ and at indicated times cells were fixed or harvested for analysis.

For Golgi disruption experiments, infected cells were washed three times with PBS followed by incubation in media containing 1 µM golgicide A (GCA) for the last 3 h of infection followed by fixation. Proteasome inhibition was performed by infecting cells for 1 h, washing the cells, and then incubating them in media containing 10 μM MG132 for 5 h. For GalNAc supplementation experiments, 20 mM of GalNAc was added to U937 cells after 1 h of infection with *L. pneumophila* and infection proceeded for 5 h prior to fixation.

### 2.4. Immunostaining and Imaging

Uninfected and *L. pneumophila*-infected cells were fixed with 4% PFA at indicated time points post infection for 20 min, followed by permeabilization using 0.1% Triton X-100 in PBS supplemented with 100 mM glycine for 20 min and blocking. All primary antibodies were diluted in PBS with 2% milk or FBS and incubated for 1 h. To label *L. pneumophila*, 1:500 mouse monoclonal antibody or 1:3000 rabbit polyclonal antibody was used. To label the Golgi, 1:200 GM130 antibody or 1:200 TGN46 antibody was used. All secondary antibody staining was carried out with Cy^TM^2-, Cy^TM^3-, or Cy^TM^5- conjugated antibodies in PBS with 1% FBS for 1 h. To label cis- and trans-Golgi lectins, 1:500 fluorescent HPA or PHA-L, respectively, was added to cells and incubated for 1 h after blocking cells with 5% FBS. To label the nucleus, cells were washed 2 times with double distilled water and incubated with 1:10,000 DAPI for 10 min. Cells were mounted using Dako Fluorescent Mounting Medium (Agilent Technologies Canada, Mississauga, ON, Canada) for confocal microscopy analysis. Z-stack images at 0.2 µm per slice were acquired at 40× or 63× using a WaveFX-X1 Spinning Disk Confocal Microscope (Quorum Technologies, Guelph, ON, Canada) and all imaging parameters including exposure, laser intensity, and gain were kept constant across samples in each trial.

For TEM analysis, *L. pneumophila*-infected macrophages on coverslips were fixed in 2% glutaraldehyde in 0.1 M Sorenson’s phosphate buffer at pH 7.2 for 2 h. Next, cells were post-fixed in 1% osmium tetroxide and 1.25% potassium ferrocyanide in sodium cacodylate buffer at room temperature for 45 min and stained for 30 min with 1% uranyl acetate in water followed by dehydration, embedding, and grid staining. Sections were imaged using an H-7500 TEM (Hitachi High Technologies, Canada, Rexdale, ON, Canada). The number of individual Golgi stacks per cell and the number of cisternae per Golgi stack were then quantified, similar to previous Golgi studies [29].

### 2.5. Immunoblotting

Uninfected and *L. pneumophila*-infected cells were washed with PBS and scraped off and lysed in cold 1x RIPA lysis buffer supplemented with a protease inhibitor cocktail at 4 °C overnight. Protein concentration was measured, and samples were mixed with loading buffer and SDS-PAGE and immunoblotting was performed. Primary antibody dilutions were: SLC35A2 1:1000, α-tubulin 1:10,000, β-actin 1:10,000, and LAMP1 1:1000. Image J was used to quantify band intensities.

### 2.6. Data Analysis and Statistics

Images were captured with MetaMorph (Molecular Devices, San Jose, CA, USA) and imported into Volocity Image Analysis (PerkinElmer, Waltham, MA, USA) or ImageJ (NIH) for Golgi analyses. To quantify the cis- or trans-Golgi area, the threshold for GM130 or TGN46 intensity was defined such that only fluorescent signal was included. The area covered by these fluorescent components and total fluorescence levels of proteins were calculated using Volocity software for 50 infected cells in 4 independent experiments. To quantify the number of *L. pneumophila* per infected macrophage, infected cells were stained for *L. pneumophila* as described above and counted visually using 63X oil-immersion objective using an inverted Zeiss epifluorescent microscope in 50 different cells from 3 independent experiments. For lectin quantifications, raw images were first processed through Z-projections using the average intensity type. The same range of interest was selected for the same protein target across all the datasets in the same sets of experiments with Golgi areas identified using immunostaining. The integrated density of the immunofluorescent signal of the selected area was then quantified. Statistical analysis was carried out using Prism from GraphPad software Inc. (La Jolla, CA, USA). A one-way ANOVA or two-way ANOVA was used followed by Tukey test for multiple comparisons. Data shown represent mean ± S.E.M. from three independent experiments unless stated otherwise. All figures were constructed using Adobe Illustrator CS6 (San Jose, CA, USA).

## 3. Results

### 3.1. Golgi Morphology and Golgi Size Remains Intact in L. pneumophila-Infected U937 Human Macrophages

*L. pneumophila* effectors in infected macrophages hijack important ER-to-Golgi trafficking regulators such as Rab1 and Arf1 in order to establish a specialized vacuole for efficient intracellular growth [6]. This, in turn, will disturb the balance between anterograde and retrograde trafficking, which we hypothesized would lead to Golgi fragmentation. In order to test this hypothesis, we performed detailed analysis of Golgi structure in *L. pneumophila*-infected macrophages using confocal microscopy. Specifically, human U937 macrophages were infected with the wild-type *L. pneumophila* Lp1 strain at an MOI of 50 and fixed at different times for subsequent immunostaining of external and total *L. pneumophila*, the cis-Golgi marker GM130, and the nuclei (Figure 1). The overall structure of the cis-Golgi remained unchanged throughout the infection and looked similar to the Golgi structure in uninfected cells (Figure 1A). To quantify the size of the Golgi compartment between non-infected and infected macrophages, Volocity software was used to measure the area covered by Golgi-positive immunostaining (Figure 1B) as well as the sum fluorescence intensity of Golgi-staining organelles in macrophages with and without *L. pneumophila* infection for varying time periods. Statistical analysis of this data revealed no significant difference in the cis-Golgi area between non-infected and *L. pneumophila*-infected U937 macrophages at any time point after infection (Figure 1B). Furthermore, there was no significant difference in the total amount of GM130 level per cell, determined by measuring the fluorescence intensity, between uninfected and infected macrophages (Figure 1C). Very similar results were seen in infected cells immunostained with TGN46 antibodies, a marker of the trans-Golgi network [30] (Appendix A), suggesting that the entire Golgi complex in human macrophages remains intact after *L. pneumophila* infection.

To definitively establish whether the Golgi structure in human macrophages was impervious to *L. pneumophila* infection, we turned to high-resolution TEM analysis. U937 macrophages were infected with *L. pneumophila* and fixed at different times after infection for subsequent TEM processing and imaging. *L. pneumophila*-infected macrophages contained typically organized Golgi stacks that were located near the nucleus, with similar numbers of cisternae observed in uninfected macrophages (Figure 2A). For detailed analysis, the number of Golgi stacks per cell was analyzed in 50 macrophages infected with or without *L. pneumophila*. The number of Golgi stacks per macrophage without and with *L. pneumophila* infection ranged from 6–8 and 4–9, respectively (Figure 2B). The number of cisternae per Golgi stack was also tabulated, from images similar to the enlarged insets in Figure 2A. The average number of cisternae per Golgi stack in both uninfected and infected macrophages was 5, with no statistically significant difference between the conditions (Figure 2C). Therefore, the structural integrity of the Golgi complex was unchanged in macrophages infected with *L. pneumophila* and was similar to the Golgi in uninfected macrophages, despite the known disruption of important ER–Golgi trafficking regulators such as Rab1 in infected cells.

### 3.2. Experimental Golgi Fragmentation Decreases L. pneumophila Growth in Human Macrophages

We wondered if *L. pneumophila* requires a proper Golgi structure for its survival, and therefore maintains the structural integrity of this complex, despite intercepting important vesicle trafficking regulators and inputs. To test this possibility, we perturbed the structure of the Golgi complex using golgicide A (GCA) and examined the subsequent effect on *L. pneumophila* growth within U937 macrophages. GCA is a highly specific and reversible inhibitor of the cis-Golgi ArfGEF, GBF1 [31] and treatment causes rapid dissociation of COPI vesicle coats from Golgi membranes, ultimately resulting in disassembly of the Golgi complex. To examine the effect of GCA on *L. pneumophila* growth, we infected U937 macrophages with *L. pneumophila* followed by GCA treatment for 3 h at early and late time points of infection (6 h and 9 h), since this is when *L. pneumophila* begins to grow and rapidly increase in number within the LCV, respectively [32]. Next, we immunostained for external and total *L. pneumophila*, cis-Golgi, and the nucleus for microscopy analysis. Uninfected macrophages treated with GCA had Golgi components that were dispersed throughout the cytoplasm compared to a more compact Golgi observed in cells without GCA (Figure 3A). This dispersed Golgi phenotype was confirmed by quantifying the area covered by cis-Golgi in GCA-treated cells, which was significantly larger compared to the area covered by cis-Golgi in untreated cells (Figure 3B). At the earlier time point of infection, GCA-induced Golgi disruption did not perturb the number of *L. pneumophila* present in macrophages (Figure 3C,E), compared to control cells. In contrast, addition of GCA at later time points resulted in significantly less bacteria in macrophages treated with GCA, compared to control cells (Figure 3D,E). Therefore, Golgi disruption induced by GCA significantly reduced *L. pneumophila* growth at later phases of infection of human macrophages.

### 3.3. L. pneumophila Infection of Macrophages Decreases Lectin Intensity for O-Glycans but Not N-Glycans

The primary function of the Golgi complex is to modify and secrete proteins and lipids that arrive from the ER. We next assessed Golgi functionality in *L. pneumophila*-infected macrophages using fluorescently tagged lectins as probes for glycosylation events [33]. *Helix pomatia agglutinin* (HPA) is a lectin that is highly specific for terminal N-Acetyl-Galactosamine (GalNAc) carbohydrate moieties and strongly labels the cis-Golgi [34]. GalNAc residues are attached to the O-groups of serine and threonine on glycoproteins with other sugar residues such as galactose, N-acetyl-Glucosamine (GluNAc), and fucose additionally added in the cis-Golgi [22]. In order to examine the glycosylation activity of Golgi in *L. pneumophila*-infected macrophages, we fixed infected macrophages at 1, 3, 6, and 10 h time points after infection for subsequent immunostaining of external and total *L. pneumophila* as well as fluorescently labelled HPA lectin. Confocal microscopy images showed a gradual increase in the number of *L. pneumophila* per infected macrophage as infection progressed, which indicated typical intracellular growth of *L. pneumophila* (Figure 4A,B). At 1 h after infection, the intensity of fluorescent HPA was similar to the HPA fluorescence intensity in uninfected macrophages (Figure 4A,D). Intriguingly, HPA fluorescence levels were so low at 3, 6, and 10 h after infection that they were largely undetectable (Figure 4A). We validated these observations by carrying out image analysis which confirmed that while GM130 levels remained similar in infected cells (Figure 4C), the intensity of HPA fluorescence significantly decreased at 3, 6, and 10 h after infection, compared to the fluorescence levels in uninfected macrophages (Figure 4D). Collectively, these experiments revealed that *L. pneumophila* infection of macrophages severely compromised glycosylation activity in the Golgi within a few hours of infection.

Given this striking lectin phenotype, we then wanted to test if N-glycosylation levels were also impacted in the host cell Golgi after *L. pneumophila* infection. Differentiated U937 human macrophages were infected with *L. pneumophila* for 1 h, 3 h, 6 h, or 10 h and fixed and stained with anti-GM130 or -TGN46 and anti-LP1 antibodies as well as Alexa Fluor^TM^488-conjugated Phaseolus vulgaris leucoagglutinin (PHA-L) lectin. PHA-L recognizes the 2,6-branched tri-tetrantennary complex-type N-glycan which contains mannose, GluNAc, and galactose [35]. PHA-L staining localized with GM130 (not shown) and strongly co-localized with TGN46 in the trans-Golgi (Figure 5A). Quantification of PHA-L staining intensity from confocal slices revealed that there was no significant difference in PHA-L staining in U937 cells infected for 3 h or 6 h compared to PHA-L in uninfected control cells (Figure 5B). A prominent N-linked glycoprotein is lysosomal-associated membrane protein 1 (LAMP1), which is heavily glycosylated with the typical mannose-6-phosphate targeting signature for lysosomal proteins [36]. We compared LAMP1 immunoblots of cell lysates from control macrophages versus cells infected with *L. pneumophila*, where band smears are indicative of glycosylation levels (Figure 5C). No significant differences were observed in LAMP1 band intensities between uninfected cells and macrophages infected with *L. pneumophila* (Figure 5D). Together, these data suggest that *L. pneumophila* infection of macrophages impacts the availability of the O-linked sugar GalNAc, but not N-linked glycoprotein-associated mannose, GluNAc, and galactose in the Golgi.

### 3.4. SLC35A2 Transporters Are Significantly Reduced in L. pneumophila-Infected Macrophages and GalNAc Supplementation Supports Bacteria Growth

One explanation for the reduced HPA lectin staining level in the Golgi could be that the availability of GalNAc in the Golgi is decreased. SLC35A2 is a UDP-galactose and UDP-GalNAc transporter that is expressed on Golgi membranes [37]. We first investigated whether the transporter levels were impacted by *L. pneumophila* infection. Using immunoblotting of total cell lysates of infected U937 and HeLa cells, we observed a marked decrease of total SLC35A2 levels within 3 h of *L. pneumophila* infection (Figure 6A,B). SLC35A2 is a large integral protein with 10 transmembrane domains [38]. Transmembrane proteins can be difficult to purify using immunoblotting [39] and we achieved more success extracting this protein in HeLa cells, which also showed HPA down-regulation in the Golgi after *L. pneumophila* infection (Appendix A).

Next, we wondered if there was a bacterial benefit to removing SLC35A2 from human macrophages. We hypothesized that reducing this transporter would reduce GalNAc movement from the cytosol to the Golgi lumen, which the bacteria could then use for its growth. To test this hypothesis, we infected U937 cells infected with *L. pneumophila* in media that were supplemented with 20 mM GalNAc. After 6 h of infection, we then stained cells with a *L. pneumophila* antibody and quantified the amount of bacteria present in infected cells. We found that the *L. pneumophila* fluorescent intensity in cells supplemented with additional GalNAc was significantly higher than cells without exogenous GalNAc treatment (Figure 6C). From these results, we conclude that GalNAc availability favors the intracellular growth of *L. pneumophila* in human macrophages.

Finally, we explored potential mechanisms driving *L. pneumophila*-induced O-glycan reduction in human macrophages by investigating the role of bacterial effectors. We infected U937 cells with *L. pneumophila* mutants in DotA and Ankyrin-repeat protein B (AnkB). DotA is part of the Dot/Icm transporter, which *L. pneumophila* uses to translocate effector proteins into host cells [3]. The ΔDotA *L. pneumophila* mutants are not virulent to host cells as they cannot inject bacterial effector proteins into host cells [40]. Infected cells were immunostained for HPA and GM130 after 3 and 6 h of infection with ΔDotA *L. pneumophila* (Figure 7A). In these cells, HPA levels were high and comparable to Golgi HPA levels in uninfected cells (Figure 7B), indicating that bacteria invasion and effector secretion were mediating the reduction in O-glycans in infected cells. We hypothesized that *L. pneumophila* may be injecting proteins that mark SLC35A2 for proteolytic degradation. AnkB is an F-box E3 ubiquitin ligase effector protein that is located on the LCV membrane and is required for the degradation of proteins from ER-derived vesicles into amino acids, which are used by *L. pneumophila* as its main energy resource [41,42].

Loss of AnkB negatively affects *L. pneumophila* intracellular growth [41]. We infected U937 macrophages with ΔAnkB *L. pneumophila* for 3 or 6 h, and still saw a significant, progressive loss in Golgi HPA, compared to HPA in uninfected cells (Figure 7C). To further confirm that proteasome activity was not responsible for the O-glycan defect in *L. pneumophila*-infected cells, we treated U937 cells with 10 μM of the proteasome inhibitor, MG132, 1 h after infection for 5 h. Cells were then stained for HPA, GM130, and *L. pneumophila*, and image intensity was quantified in infected cells. We found that MG132 treatment did not rescue the HPA loss observed in *L. pneumophila*-infected cells (Figure 7D), further confirming that SLC35A2 is likely not degraded through the ubiquitin–proteasome pathway. Together, these results indicate that *L. pneumophila* uses other strategies beyond ubiquitin–proteasome-mediated degradation to potentially remove the SLC35A2 protein in host cells and impair O-linked glycosylation events.

## 4. Discussion

In this study, we determined that despite known disruption of ER–Golgi trafficking, *L. pneumophila* infection of human macrophages does not affect the ultrastructure, morphology, and size of the Golgi. One way to explain this is that the Golgi to plasma membrane and Golgi to lysosome trafficking are also dampened to restrain the Golgi volume from decreasing, or that there is a sufficient basal delivery of ER-derived and retromer vesicles to accommodate the Golgi volume. A recent paper on Cos7 and A549 cells examined potential biosynthetic trafficking defects caused by *L. pneumophila* by monitoring VSVG movement through the secretory pathway [43]. There was a significant disruption in anterograde trafficking through the secretory pathway, however, *L. pneumophila*-induced Golgi fragmentation occurred in these cell types, through SdeA-mediated degradation of the Golgi structural proteins, GRASP55 and GRASP65 [43]. We are currently investigating trafficking flow through the intact Golgi in *L. pneumophila*-infected macrophages to determine whether there is an impact on the biosynthetic pathway.

We also showed in this study that an intact Golgi is required for *L. pneumophila* growth in human macrophages. This is in contrast to other pathogenic bacteria that disrupt the Golgi. Several bacteria manipulate host cell Golgi structure and dynamics to generate an environment that favors their intracellular replication. *Chlamydia trachomatis*, *Orientia tsutsugamushi*, and *Anaplasma phagocytophilum* infections hijack host cell Golgi-derived vesicles [44,45,46]. *Shigella* withdraws cholesterol from the plasma membrane which impairs ER to Golgi to plasma membrane forward trafficking as well as plasma membrane to Golgi retrograde trafficking, leading to Golgi fragmentation without affecting ER morphology [47,48]. While PI(4)P, normally found on the Golgi membrane, is enriched on the LCV membrane [49,50], other Golgi proteins are not [43,51,52], suggesting that vesicles from this organelle are not co-opted during *L. pneumophila* infection. Golgi stressors are known to activate CAMP Responsive Element Binding Protein 3 (CREB3) and ADP-ribosylation factor 4 (ARF4) to preserve the Golgi integrity [47,53]. Inhibiting the CREB3-ARF4 pathway increases host cell resistance to *C. trachomatis* and *Shigella* infections [53]. It will be interesting to see if *L. pneumophila* induces similar signaling to maintain an intact Golgi and to prevent activation of Golgi stress pathways until the very late stages of infection.

While the Golgi structure persists in *L. pneumophila*-infected macrophages, the function of the Golgi is seemingly impaired, with a notable reduction in O-glycans. The reduction in Golgi HPA-lectin staining as a proxy for O-glycosylation events in the Golgi is dependent on the DotA effector expression but not AnkB expression during *L. pneumophila* infection. Strikingly, bacterial infection of human macrophages induced a global reduction in levels of the UDP-GalNAc transporter, SLC35A2. Intriguingly, a recent genome-wide CRISPR screen identified SLC35A2 as a factor that impacts *L. pneumophila* intracellular growth within U937 macrophages. Knocking down SLC35A2, but not other SLC35A family proteins, increased the pathogenesis of *L. pneumophila* in U937 cells [54], supporting our findings. SLC35A2 is also an important transporter for UDP-galactose across the Golgi membrane [37] (however, we did not detect measurable reductions in galactose-containing N-glycan levels, as detected by PHA-L staining and LAMP1 immunoblotting of *L. pneumophila*-infected macrophages. We are currently conducting mass spectrometry analysis for a comprehensive study of macrophage glycans impacted by *L. pneumophila* infection and investigating the fidelity of other galactose transporters in the ER and Golgi.

We originally hypothesized that the reduction in intracellular SLC35A2 after *L. pneumophila* infection is through proteasome degradation. *L. pneumophila* translocates many effector E3 ligase proteins, including AnkB, into the host cell cytosol to degrade proteins into amino acids for their utilization as an energy resource [55]. However, we still detected attenuated HPA lectin staining in the Golgi of U937 macrophages infected with ΔAnkB *L. pneumophila*. One possibility is that SLC35A2-containing vesicles are recruited to the LCV from the ER and polyubiquitinated by E3 ligases other than AnkB on the LCV. Another possibility is that *L. pneumophila* E3 ubiquitin ligase effector GobX is translocated specifically onto Golgi membranes through S-palmitoylation [56] to mediate degradation of SLC35A2. However, in our hands, proteasome inhibition did not rescue HPA lectin Golgi levels in *L. pneumophila*-infected macrophages, suggesting this was not the degradation mechanism. It is possible that other lysosome-mediated degradation pathways, including microautophagy or chaperone-mediated autophagy [57] are important for the reduction in SLC35A2 and Golgi O-glycans in infected macrophages, and this remains an open avenue of investigation.

We postulate that *L. pneumophila*-induced degradation of the SLC35A2 transporter is the main underlying mechanism that accounts for the loss of GalNAc on glycoproteins. SLC35A2 was difficult to routinely purify from the human U937 macrophage-like cell line, so we turned to HeLa cells for these analyses. HeLa cells do not possess phagocytic properties; however, HeLa cells have been used for decades to study *L. pneumophila* virulence [58]. Indeed, we observed a very similar phenotype of intact Golgi with reduced HPA reactivity in HeLa cells infected with *L. pneumophila*, confirming these cells as a suitable model to study biosynthetic pathway modulations by *L. pneumophila*. Given our results, it cannot be ruled out that *L. pneumophila* may negatively affect the O-glycosylation process through inhibiting the function, delivery, or expression of Golgi GalNAc transferases. Although GalNAc transferases have 20 family members [59], of particular interest are the enzymes that directly catalyze the GalNAc attachment onto the O-residue of serine/threonine residue of glycoproteins, which is the major substrate of the HPA lectin we utilized in this study.

It remains a possibility that *L. pneumophila* is actively targeting O-glycans to modify the host cell surface. Glycans are the first point of contact for microbes, and they must penetrate O-linked mucins in mucous and the glycocalyx oligosaccharides to allow successful pathogen binding and cell entry. For instance, *S*. Typhimurium secretes chitinases and sialidases that target N-linked GlcNAc-containing glycans and sialic acid, respectively [60,61,62,63]. Gut microbes frequently target O-glycans such as mucin to disrupt mucous viscosity or to use as an energy source [64]. These examples are extracellular pathogen strategies; however, our study revealed an intracellular modulation of host cell glycosylation by *L. pneumophila*. Interestingly, host cell infection by *S*. Typhimurium also triggers host cell upregulation of N-linked surface glycans [60], perhaps as a defensive response.

In bacteria, GalNAc is used for the synthesis of the cell wall and is a component of lipopolysaccharides [65]. GalNAc is also an energy resource for bacteria such as *E. coli* [65]. *L. pneumophila* primarily uses amino acids as its main energy resource [66] However, *L. pneumophila* is also fueled by host cell glucose, which is required for the generation of its cell wall components and for its intracellular replication within eukaryotic cells [67]. Here, we show that supplementing human macrophages with GalNAc supported *L. pneumophila* growth. These observations are novel and suggest an important role for sugars for *L. pneumophila* growth within macrophages. Future work should determine whether comparable GalNAc transporters exist on the LCV to co-opt this host sugar and how the bacteria processes and exploits GalNAc for its infectious cycle.

## Figures and Tables

**Figure 1 pathogens-11-00908-f001:**
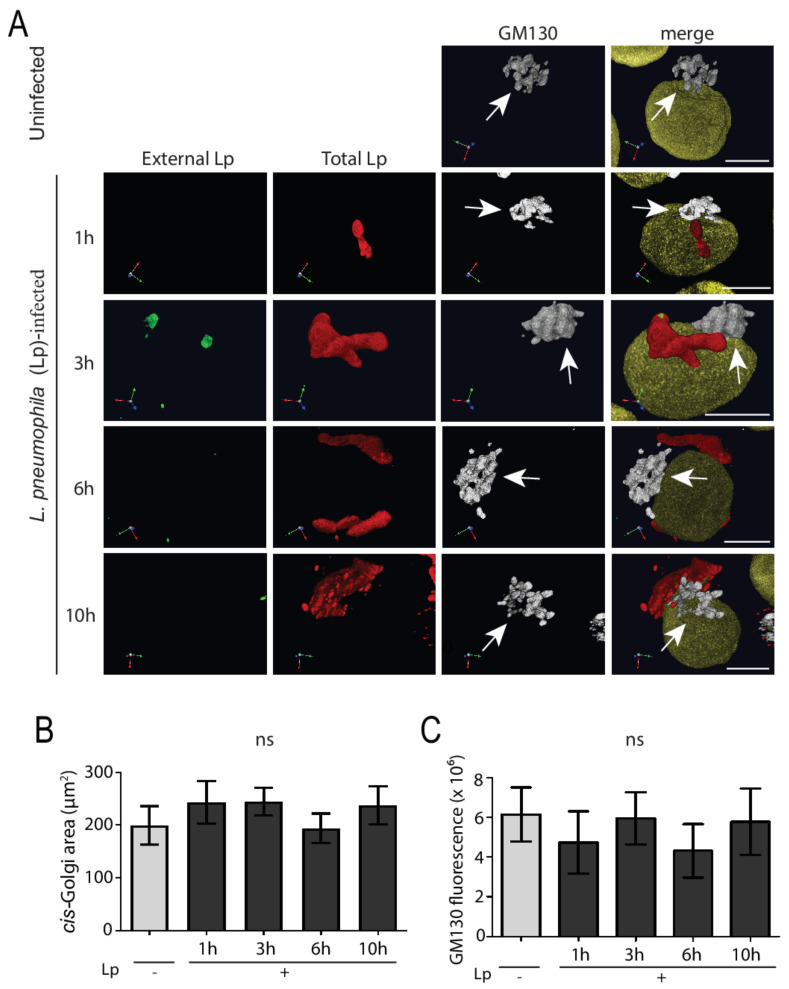
The cis-Golgi structure remains intact in *L. pneumophila*-infected U937 macrophages. (**A**) U937 macrophages infected with *L. pneumophila* were fixed at indicated time points followed by immunostaining to visualize external *L. pneumophila* (green), total *L. pneumophila* (red), the cis-Golgi marker GM130 (white), and the nucleus (yellow). Arrows point to intact cis-Golgi structures. Scale bars = 5 µm. Quantifications of (**B**) cis-Golgi area and (**C**) total GM10 fluorescence in uninfected (Lp -) and infected (Lp +) cells at each time point. ‘Lp’ refers to *L. pneumophila*. Data represent mean ± S.E.M. from four independent experiments (n = 50, ns = not significant, one-way ANOVA test).

**Figure 2 pathogens-11-00908-f002:**
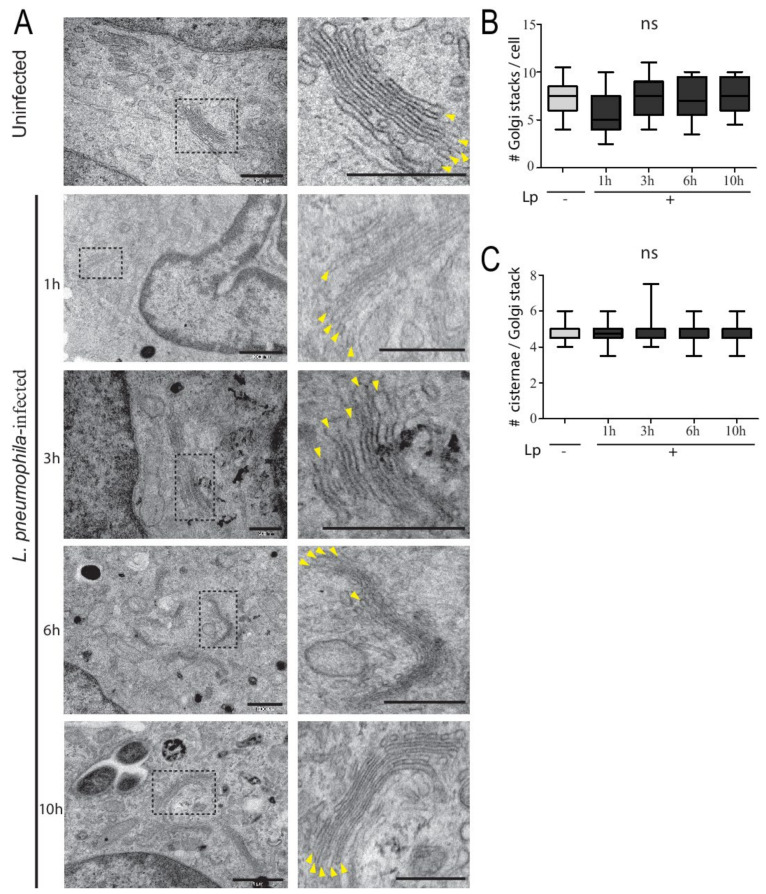
U937 macrophages infected with *L. pneumophila* have typical Golgi complex ultrastructure compared to uninfected cells. (**A**) TEM of Golgi stacks in uninfected and infected U937 macrophages at different time points post-infection. Panels on right represent zoomed images of dashed region in original image. Yellow arrowheads in zoomed images point to individual cisternae within the Golgi stack. Scale bars = 500 nm. (**B**) Number of Golgi stacks per cell in uninfected (Lp -) and infected (Lp +) macrophages at different time points of infection. (**C**) Number of cisternae per Golgi stack in uninfected (Lp -) and infected cells (Lp +) macrophages at different time points of infection. Data represent mean ± S.E.M. from two independent experiments (n = 25, ns = not significant, one-way ANOVA test).

**Figure 3 pathogens-11-00908-f003:**
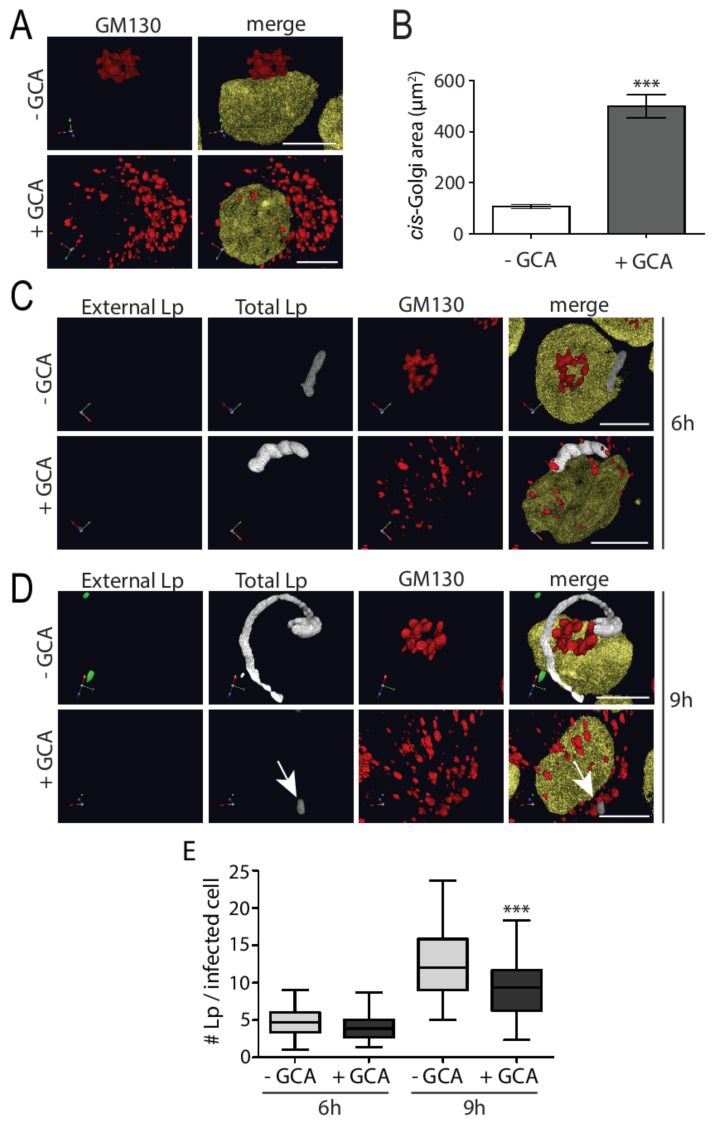
Golgicide-A-induced Golgi disruption significantly decreases multiplication of *L. pneumophila* within U937 macrophages at late phase of infection. (**A**) U937 macrophages were infected with *L. pneumophila* for 6 or 9 h in the presence or absence of 1 µM golgicide A (GCA) for the last 3 h of infections. Cells were fixed and immunostained to visualize the cis-Golgi marker GM130 (red) and nuclei (yellow). (**B**) Quantification of cis-Golgi area in cells without (- GCA) and with golgicide A (+ GCA) to show the area over which the Golgi dispersed with GCA treatment. *L. pneumophila*-infected U937 macrophages treated with or without 1 µM GCA followed by fixing and staining at (**C**) 6 h and (**D**) 9 h post-infection to visualize external *L. pneumophila* (green), total *L. pneumophila* (white), the cis-Golgi marker GM130 (red), and nucleus (yellow). Arrow indicates reduced *L. pneumophila* growth in GCA-treated macrophages. Scale bars = 5 µm. (**E**) Graph representing number of *L. pneumophila* per infected macrophage in U937 cells treated with or without GCA after 6 h and 9 h infection. Data represent mean ± S.E.M. from three independent experiments (n = 50, *** *p* = 0.0001, two-way ANOVA test).

**Figure 4 pathogens-11-00908-f004:**
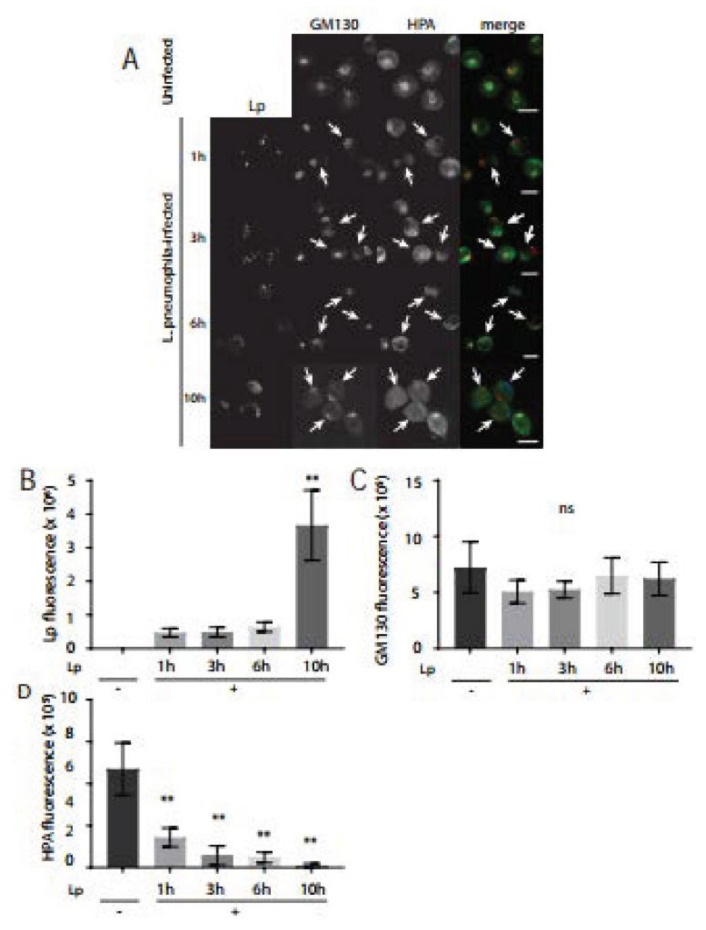
HPA lectin staining levels significantly decrease in *L. pneumophila*-infected U937 macrophages compared to uninfected macrophages. (**A**) U937 macrophages infected with *L. pneumophila* were fixed at indicated time points followed by immunostaining to visualize *L. pneumophila* (blue), GM130 (red), and the cis-Golgi lectin, *Helix pomatia* agglutinin (HPA) (green). Arrows indicate reduced HPA lectin fluorescence signal in *L. pneumophila*-infected cells. Scale bars = 10 µm. Quantifications of (**B**) *L. pneumophila* fluorescence, (**C**) GM130 fluorescence, and (**D**) HPA lectin fluorescence levels in uninfected (Lp -) and infected (Lp +) cells at each time point. Data represent mean ± S.E.M. from three independent experiments (n = 18, ** *p* < 0.01, ns = not significant, one-way ANOVA test, Tukey test).

**Figure 5 pathogens-11-00908-f005:**
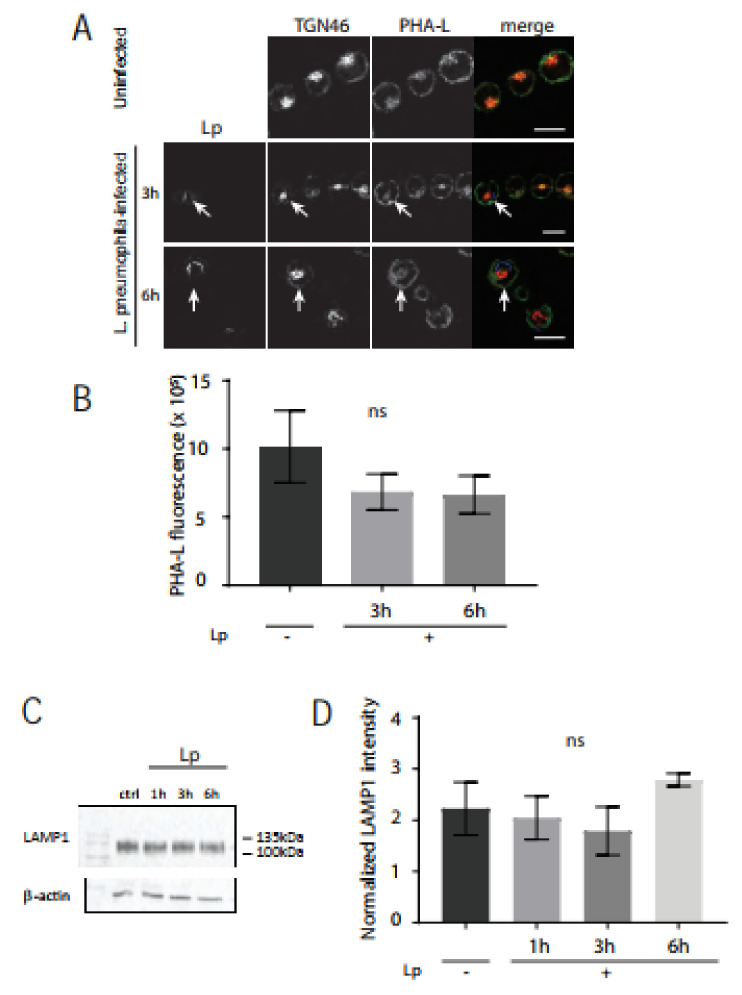
PHA-L lectin and LAMP1 expression levels are unchanged in *L. pneumophila*-infected U937 macrophages, compared to uninfected macrophages. (**A**) U937 macrophages infected with *L. pneumophila* were fixed at indicated time points followed by immunostaining to visualize *L. pneumophila* (blue), TGN46 (red), and the lectin phytohemagglutinin-L (PHA-L) (green). Arrows indicate PHA-L lectin fluorescence signal in *L. pneumophila*-infected cells. Scale bars = 10 µm. (**B**) Quantification of PHA-L lectin fluorescence levels in uninfected (Lp -) and infected (Lp +) cells at each time point. Data represent mean ± S.E.M. from three independent experiments (n = 12, ns = not significant, one-way ANOVA test, Tukey test). (**C**) LAMP1 immunoblotting of cell lysates from U937 cells infected with *L. pneumophila* for indicated time points. Cell lysates were probed with an anti-β-actin antibody as a loading control. (**D**) Densitometry analysis of 3 replicate blots where LAMP1 levels were normalized to β-actin: ns = not significant using one-way ANOVA, Tukey test.

**Figure 6 pathogens-11-00908-f006:**
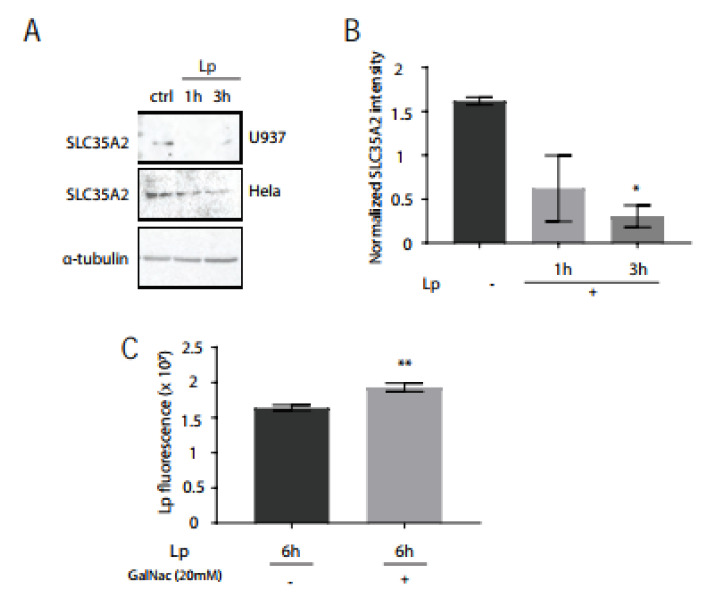
*L. pneumophila* infection reduces total SLC35 protein levels and GalNAc supplementation promotes *L. pneumophila* growth in human macrophages. (**A**) Immunoblotting for SLC35A2 in U937 and HeLa cells infected with *L. pneumophila*. Cell lysates were probed with an anti-α-tubulin antibody as a loading control. (**B**) Densitometry results of 1 blot of U937 cells and 2 replicate HeLa cell blots, where a significant reduction in signal was observed in *L. pneumophila*-infected cells compared to control cells. * *p* < 0.05, one-way ANOVA, Tukey test. (**C**) Quantification of *L. pneumophila* fluorescence intensity after infection of U937 cells for 6 h, with the presence or absence of supplemental 20 mM GalNAc for the last 5 h of infection, prior to fixing and immunostaining with *L. pneumophila* antibodies. ** *p* < 0.01, n = 51 from three independent experiments, one-way ANOVA, Tukey test. Data represent mean ± S.E.M.

**Figure 7 pathogens-11-00908-f007:**
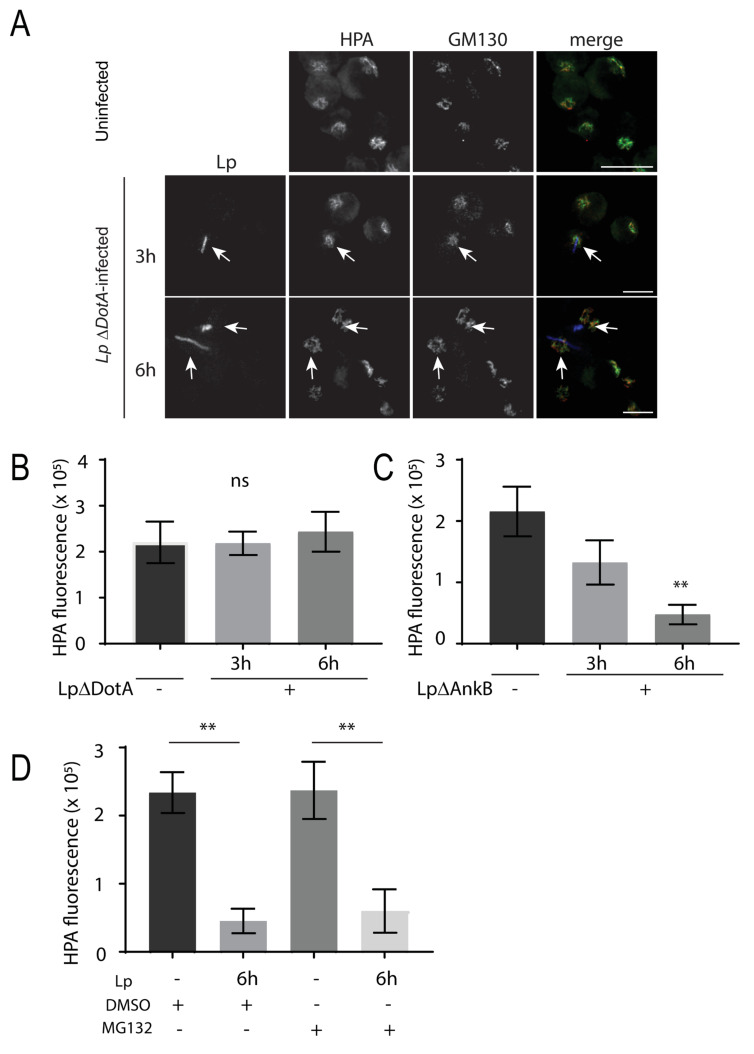
*L. pneumophila* infection requires DotA expression for reduced HPA lectin staining in the Golgi, but not AnkB or host cell proteasome activity. (**A**) U937 macrophages infected with *L. pneumophila* ∆DotA were fixed at indicated time points followed by immunostaining to visualize *L. pneumophila* (blue), GM130 (red), and HPA (green). Arrows indicate HPA lectin fluorescence signal in *L. pneumophila* ∆DotA-infected cells. Scale bars = 10 µm. (**B**) Quantifications of HPA lectin fluorescence levels in the Golgi in uninfected (Lp ∆DotA -) and infected (Lp ∆DotA +) cells at each time point. (**C**) Quantifications of similar experiments where U937 cells were infected with *L. pneumophila* ∆AnkB mutants followed by fixing and staining for the HPA lectin and the Golgi. HPA fluorescence levels are shown from uninfected (Lp ∆AnkB -) and infected (Lp ∆AnkB +) cells at each time point. Data represent mean ± S.E.M. from three independent experiments (n = 18, ** *p* < 0.01, one-way ANOVA test, Tukey test). (**D**) U937 cells were infected with or without *L. pneumophila* for 1 h. Cells were then treated with 10 μm MG132 or DMSO for 5 h followed by fixation and immunostaining for *L. pneumophila*, GM130, and HPA. Data show HPA fluorescent intensity quantification from confocal images from three independent experiments: n = 13, ** *p* < 0.01, one-way ANOVA, Tukey test. Data represent mean ± S.E.M.

## Data Availability

The data presented in this study are available in this article and Appendix A.

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
