# Peer review of "Legionella pneumophila* Infection of Human Macrophages Retains Golgi Structure but Reduces O-Glycans"

_pathogens, 2022, doi:10.3390/pathogens11080908_

Round 1
Reviewer 1 Report
The manuscript by Fu et al. entitled "Legionella pneumophila infection of human macrophage retains Golgi structure but reduces O-glycans" is a novel set of results examining the impact of Legionella pneumophila on the functioning of Golgi in two cell lines.
There are a few typographical errors such as the use of "L. pneu" on page 2, a full stop rather than a comma after WT in the methods on page 3, "we these data" on page 10, hijacks (should be plural) on page 15, and capitalisation/lack of italisation for typhimurium (twice) on page 16.
The layout of the figures vs. text needs to be addressed - there is too much white space in several places and Figure 6 is badly laid out and the contrast/image quality unconvincing for the blots.
My major criticism relates to the complete lack of information about the strains of Legionella used. What strain is it? Is it available from ATCC? What ST is it? Insufficient details about the recombinant strains is also given - are they the same ST as the WT? How were the knock-outs generated (the source is not a co-author and there is no citation of the studies validating/proving that dotA and ankB were actually deleted). It is therefore difficult to assess whether the model has any design flaws that could impact the interpretation of results.
Author Response
Reviewer 1 comments:
- The manuscript by Fu et al. entitled "Legionella pneumophila infection of human macrophage retains Golgi structure but reduces O-glycans" is a novel set of results examining the impact of Legionella pneumophila on the functioning of Golgi in two cell lines.
There are a few typographical errors such as the use of "L. pneu" on page 2, a full stop rather than a comma after WT in the methods on page 3, "we these data" on page 10, hijacks (should be plural) on page 15, and capitalisation/lack of italisation for typhimurium (twice) on page 16.
We apologize for the spelling and syntax errors and have corrected these in the revised manuscript. Typhimurium is the serovar of Salmonella enterica Typhimurium and is typically not italicized and thus represented as S. Typhimurium.
The layout of the figures vs. text needs to be addressed - there is too much white space in several places and Figure 6 is badly laid out and the contrast/image quality unconvincing for the blots.
We used standard portrait/letter-sized illustrator template files to assemble the figures but agree that there is excess white space between panels in some of these figures. We have reorganized the figures and placed some into landscape templates to minimize white space within the figures (eg. Figures 5, 6, Supplemental Fig. 2). We have included the highest resolution images of the blots in Figure 6 and the three lanes are now better resolved. We struggled with purifying SLC35 in macrophages, even in control cells, which is why we also included HeLa cells for these analyses, as described in the original manuscript submission. All of the revised figures have been uploaded for the latest submission.
- My major criticism relates to the complete lack of information about the strains of Legionella used. What strain is it? Is it available from ATCC? What ST is it? Insufficient details about the recombinant strains is also given - are they the same ST as the WT? How were the knock-outs generated (the source is not a co-author and there is no citation of the studies validating/proving that dotA and ankB were actually deleted). It is therefore difficult to assess whether the model has any design flaws that could impact the interpretation of results.
We regret this oversight. We used the L. pneumophila LP02 strain and we received the strains from Drs. Alexander Ensminger (University of Toronto, CAN) and Ralph Isberg (Tufts University Medical School, USA). These mutant bacteria were validated and characterized within publications that we now include in the Methods section on page 3 and below: (Berger and Isberg, 1993; Berger et al., 1994; Ensminger and Isberg, 2010; Prashar et al., 2018).
References for Reviewers
Berger, KH, and Isberg, RR (1993). Two distinct defects in intracellular growth complemented by a single genetic locus in Legionella pneumophila. Molecular Microbiology 7, 7–19.
Berger, KH, Merriam, JJ, and Isberg, RR (1994). Altered intracellular targeting properties associated with mutations in the Legionella pneumophila dotA gene. Molecular Microbiology 14, 809–822.
Ensminger, AW, and Isberg, RR (2010). E3 Ubiquitin Ligase Activity and Targeting of BAT3 by Multiple Legionella pneumophila Translocated Substrates. Infection and Immunity 78, 3905.
Prashar, A, Ortiz, ME, Lucarelli, S, Barker, E, Tabatabeiyazdi, Z, Shamoun, F, Raju, D, Antonescu, C, Guyard, C, and Terebiznik, MR (2018). Small Rho GTPases and the effector VipA mediate the invasion of epithelial cells by filamentous Legionella pneumophila. Frontiers in Cellular and Infection Microbiology 8, 133.
Reviewer 2 Report
The paper was written properly. I have only one suggestion to describe what was the purpose of the research.
Author Response
Reviewer 2 comments:
1. The paper was written properly. I have only one suggestion to describe what was the purpose of the research.
We thank the referee for this important suggestion and have now included a study purpose within the Introduction on page 2.
Reviewer 3 Report
much of this work was done with the U937 cell line, you also used the Hela cell line but the results of this cell line are neither described in the abstract section nor in the discussion section. you have two options, either remove the results about the Hela line, or present a discussion about the results of this cell line and comparing with the U937 line.
in Material and Methods - section Reagents and antibodies, are incomplete, later in the paper in the "Immunostaining and imaging" sub-section and "Immunoblotting" sub-section you continue to mention antibodies.
Author Response
Reviewer 3 comments:
- Much of this work was done with the U937 cell line, you also used the Hela cell line but the results of this cell line are neither described in the abstract section nor in the discussion section. you have two options, either remove the results about the Hela line, or present a discussion about the results of this cell line and comparing with the U937 line.
We agree that the manuscript is improved with a comparison of the two cell lines, which we have now included in the Discussion on page 15. We felt that it was worthwhile keeping the HeLa results in the manuscript as SLC35 was more readily purified in HeLa cells, compared to macrophages.
2. In Material and Methods - section Reagents and antibodies, are incomplete, later in the paper in the "Immunostaining and imaging" sub-section and "Immunoblotting" sub-section you continue to mention antibodies.
We thank the reviewer for noticing this and have now included the missing LAMP1 antibody source as well as including that the tubulin and actin antibodies were from Sigma. We revisited the antibodies in the immunofluorescence / immunoblotting sections to report their respective dilutions within each protocol.